

# The new CNES-CLS18 Global Mean Dynamic Topography

Sandrine Mulet[1], Marie-Hélène Rio[1,2], Hélène Etienne[1], Camilia Artana[3,4], Mathilde Cancet[5], Gérald Dibarboure[6], Hui Feng[7], Romain Husson[8], Nicolas Picot[6], Christine Provost[3], P. Ted Strub[9]

[1]CLS, Altimetry and In-Situ Oceanography, Ramonville Saint Agne, 31520, France

[2]ESA/ESRIN, Frascati, 00044, Italy

[3]Laboratoire LOCEAN-IPSL, Sorbonne Université (UPMC, University Paris 6), CNRS, IRD, MNHN, Paris, 75000, France

[4]Mercator Océan, Ramonville Saint Agne, 31520, France

[5]Noveltis, Labège, F-31670, France

[6]CNES, Ramonville Saint Agne, 31520, France

[7]Ocean Process Analysis Lab, University of New Hampshire, Durham, NH 03824, USA

[8]CLS Brest, Plouzané, 29280, France

[9]College of Earth, Ocean and Atmospheric Sciences, Oregon State University, Corvallis, OR 97331-5503, USA

**Abstract:** The Mean Dynamic Topography (MDT) is a key reference surface for altimetry. It is needed for the calculation of the ocean absolute dynamic topography, and under the geostrophic approximation, the estimation of surface currents. CNES-CLS Mean Dynamic Topography (MDT) solutions are calculated by merging information from altimeter data, GRACE and GOCE gravity field and oceanographic in-situ measurements (drifting buoy velocities, hydrological profiles). The objective of this paper is to present the newly updated CNES-CLS18 MDT. The main improvement compared to the previous CNES-

CLS13 solution is the use of updated input datasets: the GOCO05S geoid model is used based on the complete GOCE mission (Nov 2009-October 2013) and 10.5 years of GRACE data, together with all drifting buoy velocities (SVP-type and Argo floats) and hydrological profiles (CORA database) available from 1993 to 2017 (instead of 1993-2012). The new solution also benefits from improved data processing - in particular a new wind-driven current model has been developed to extract the geostrophic component from the buoy velocities; and methodology - in particular the computation of the medium

scale GOCE-based MDT first guess and the correlation scales used for the multivariate mapping have been revised. An evaluation of the new solution compared to the previous version and to other existing MDT show significant improvements both in strong currents and coastal areas.





# 1 Introduction

The estimation of an accurate Mean Dynamic Topography (MDT) has been a long standing issue for the reconstruction of
the absolute dynamic topography from altimeter data (Rio et al., 2010). The lack of an accurate geoid at spatial scales
corresponding to the along-track spatial resolution of altimeter data (7km at 1Hz, 300m at 20Hz) has led to the exploitation
of the time variable part of the sea level with respect to the sea level mean over a given reference period: the Sea Level
Anomaly (SLA). Consequently, for years, the use of altimetry for scientific ocean studies has focussed on the analysis of Sea
Level Anomalies. While providing invaluable insight into the ocean dynamics of mesoscale eddies, a large number of
scientific and operational activities rely on the accurate estimate of the absolute sea level. For instance, Pegliasco et al.
(2020) show that eddies are better tracked and studied using the absolute sea level instead of the sea level anomalies. Also,
the MDT is the missing component for the optimal assimilation of altimeter data into operational ocean systems, such as
those run under the Copernicus Marine Environment Monitoring Services (CMEMS). Most importantly, the absolute
dynamic topography is directly linked, under the geostrophic assumption, to the ocean surface currents. Those are required
for a wide range of applications, such as the management of fishery resources, the monitoring of potential pollution,
maritime security, etc.

Over the years, the MDT has benefitted from a number of important improvements, the major one being the measurement of
the Earth geoid at 100 km resolution with centimetric accuracy thanks to the launch in 2009 of the first ESA Earth Explorer,
GOCE (Gravity and Ocean Circulation Experiment). Another key element has been the accumulation and improved
processing over years of altimeter data leading to significant improvements in the Mean Sea Surface (MSS, Pujol et al,
2018). At scales larger than 100km, the Mean Dynamic Topography can indeed be estimated by differencing both surfaces
(MSS and geoid) and applying the adapted filter. This method is usually referred to as the "geodetic" approach (Bingham et
al, 2008). To estimate the MDT spatial scales shorter than 100km, other information is needed. The method used to calculate
the CNES-CLS MDT series (Rio and Hernandez, 2004, Rio et al, 2011, Rio et al, 2014a) is based on the merging of a
geodetic MDT with oceanic in-situ data (temperature and salinity profiles and surface drifters), processed to extract the
useful information (full dynamic height from the hydrological profiles, and geostrophic component of the drifter derived
velocities).

This paper presents the latest CNES-CLS18 MDT solution. The method is reviewed in section 2, while data used in the
computation are presented in section 3. The geodetic MDT used as first guess is described and validated in section 4. The
processing of surface drifters has been greatly improved: section 5 explains the full processing including the new wind
slippage correction and the new wind-driven current estimates, both used to remove the ageostrophic component of the
drifter currents. Section 6 describes the processing of in-situ measurements of dynamic heights before being merged with the
first guess and processed surface drifters through objective analysis (section 7). The accumulation of in-situ measurements
from different international programs, such as WOCE (for the SVP drifter) and ARGO (for the hydrological profiles) allows
us today for the first time to map the global MDT at 1/8 degree resolution (instead of ¼ degree for previous MDTs), and to



significantly improve its accuracy in coastal areas, as it will be highlighted in the validation section (section 8). Conclusions and recommendations for future work are provided in section 9.

**2 Method**

The method used to calculate the CNES-CLS18 Mean Dynamic Topography is similar to the one used in previous CNES-CLS MDT versions. A detailed description can be found in (Rio and Hernandez, 2004, Rio et al, 2011, Rio et al, 2014a). It is a three step approach.

A first MDT solution is calculated from the optimally filtered differences between an altimeter MSS and a geoid model. The effective resolution of the obtained field depends on the level of noise of the raw difference between MSS and geoid height. It thus depends on the areas, but it is around 100-125km (Bruinsma et al., 2014).

In the second step of the method, synthetic estimates of the MDT and mean geostrophic velocities are calculated using in-situ measurements of the ocean dynamic heights and surface velocities. First the in-situ measurements are processed so as to extract the geostrophic component only from the drifting buoy total velocities, and to complete the dynamic heights with the missing barotropic and deep baroclinic components. The temporal variability of the measured heights and velocities is further removed by subtracting the altimeter sea level and geostrophic velocity anomalies respectively. The processed in-situ measurements are further averaged into 1/4 and 1/8 boxes to obtain respectively the synthetic mean heights and velocities. These are finally used in the third step to improve the accuracy of the filtered MDT obtained in step 1 and add information at shorter scales. This is done through a multivariate objective analysis whose required inputs are: the synthetic mean heights and velocities and their error, the first guess MDT, the a-priori knowledge of the MDT variance and zonal and meridional correlation scales.

**3 Data**

The CNES-CLS18 MDT is calculated from a combination of altimeter and space gravity data, in-situ measurements and model winds. The following datasets are used:

- MSS: the CNES-CLS15 MSS derived for the 1993-2012 time period by Pujol et al (2018) is used.
- Geoid model: The geoid model GOCO05s (Mayer-Gürr,et al. 2015) is used with the CNES-CLS15 MSS in the computation of the MDT first guess.
- Altimeter Sea Level Anomalies (SLA): the DUACS-2018 (Taburet et al, 2019) multimission gridded sea level and derived geostrophic velocity anomaly products distributed by the Sea Level Thematic Assembly Center (SL-TAC) from CMEMS altimeter are used
- The dynamic heights are calculated from Temperature and Salinity (T/S) profiles from CORA4.2 (1993-2013), CORA5.0 (2014-2015) and CORA5.1 (2016) datasets (Cabanes et al., 2014; Szekely et al., 2019), processed by the





IN-Situ Thematic Assembly Center (INS-TAC) of the Copernicus Marine Environment and Monitoring Service (CMEMS).

-   In-situ velocities: Two types of in-situ drifting buoy velocities are used, the 6-hourly SVP type drifter distributed by the Surface Drifter Data Assembly Center (SD-DAC) and the Argo floats surface velocities from the regularly

updated YOMAHA07 dataset for the period 1993-2016  (Lebedev et al, 2007). SVP-type drifters consist of a spherical buoy with a drogue attached in order to minimize the direct wind slippage and follow the ocean currents at a nominal 15m depth. When the drogue gets lost, the drifter is then advected by the surface currents and the direct action of the wind. In this study both the 15m-drogued and the surface undrogued drifter velocities over the 1993-2016 period are considered (Lumpkin et al, 2013) for the MDT calculation. Year 2017 is used for independent

validation of the results.

-   Wind data: Wind stress data are needed for the calculation of the Wind driven velocities (section 0) that is used to remove part of the ageostrophic component from drifter velocities. We use the 3-hourly, 80 km resolution wind stress fields from ERA-Interim (Dee et al, 2011) for the period 1993-2017.

-   Mixed Layer Depth (MLD): In the computation of the wind driven velocities, an estimation of the MLD is also

needed. We use the delay time 3D temperature and salinity fields from ARMOR3D, computed by the Multi Observation Thematic Assembly Centre (MOB-TAC) of CMEMS (Guinehut et al, 2012).

## 4 First Guess calculation and comparison with previous first guess

The raw difference between the CNES-CLS15 MSS and GOCO05s geoid height is filtered using an updated version of the optimal filter fully described in Rio et al. (2011). The a priori statistics (errors and variance) are estimated using the MDT

CNES-CLS13 (Rio et al., 2014a). The correlation radii ($x_0$, $y_0$) are the same as those used in the computation of the first guess of MDT CNES-CLS13 (Rio et al., 2014).

The mean geostrophic currents associated with the resulting first guess are compared with mean geostrophic currents estimated from drifters (section 0) and filtered at 80 km of resolution (the smallest scales that GOCE is able to resolve considering its lowest orbit at the end of the mission). Comparison is done outside of the equatorial band [-5; 5] where the

geostrophic approximation is not valid. The Root Mean Square Difference (RMSD) for the zonal component is almost everywhere lower than 30% of the RMS of the drifters (Figure 1a). The RMSD is higher for the meridional component (RMSDV) since the signal is smaller and thus more difficult to extract from both geodetic MDT and drifters (smaller signal over noise ratio). RMSDV is often less than 60% but reaches 100% in the middle of the Pacific (Figure 1b). Figure 2 illustrates the improvement compared with the previous first guess estimated for the MDT CNES-CLS13. The two plots

show the differences between RMSD of the MDT CNES-CLS13 first guess and RMSD of the MDT CNES-CLS18 first guess for both components of the mean geostrophic velocities. The RMSD is reduced almost everywhere (reddish colors).





For instance, both components are improved along the coast in the Gulf of Maine and the meridional component is improved along the Chilean coast (the improvement in these areas is further described in sections 0 and 0 respectively).

a)                                                                      b)

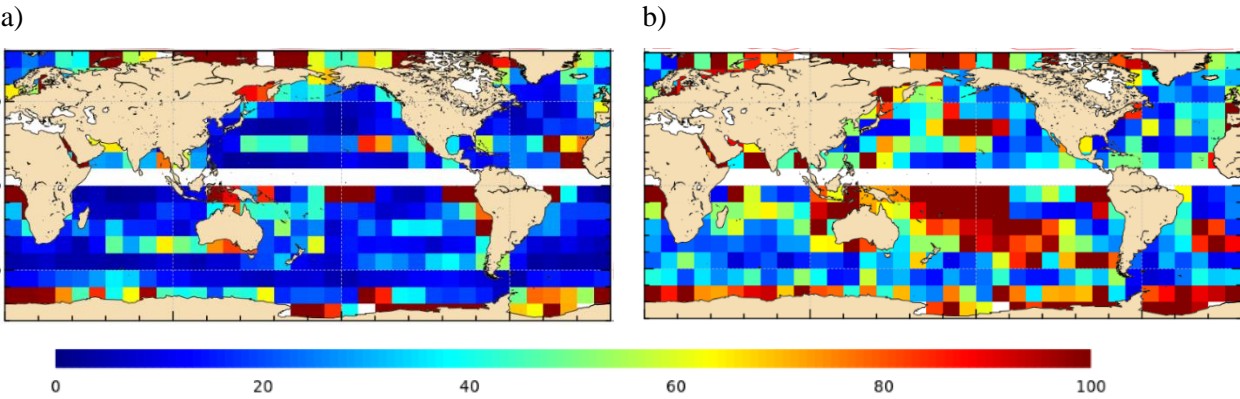

**Figure 1: Root Mean Square differences between (a) zonal and (b) meridional mean geostrophic drifter velocities and the first guess geostrophic velocities in percentage of drifter variance (%) in 5°x5° boxes**

a)                                                                      b)

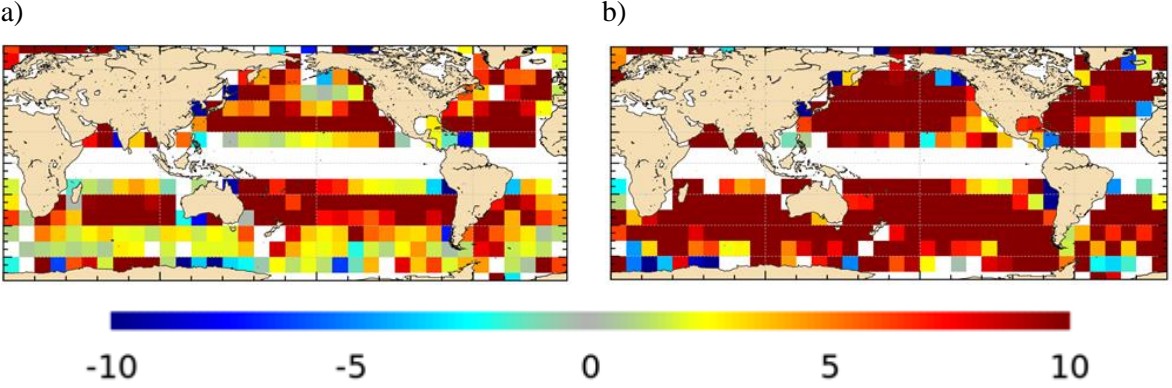

**Figure 2: Difference (cm/s) of the RMSD obtained using the first guess of MDT CNES-CLS13 and the first guess of MDT CNES_CLS18 for the (a) zonal and (b) meridional components in 5°x5° boxes. Only boxes with more than 500 points and where difference is statistically significant at more than 95% (T-test) are shown. The reddish color underline improvement while bluish stand for degradation (see text for more detail)**

## 5 Drifter data processing

As in previous work (Rio and Hernandez, 2004, Rio et al, 2011, 2014a), the mean synthetic velocities $\vec{U}_{synth}$ are calculated by removing from the drifting buoy velocities $\vec{U}_{buoy}$ both the ageostrophic velocity components and the temporal variability $\vec{U}'_{alti}$ of the geostrophic velocity components as measured by altimetry (Eq. 1)



$$\vec{U}_{synth} = \vec{U}_{buoy} - \vec{U}_{ekman} - \vec{U}_{stokes} - \vec{U}_{inertial} - \vec{U}_{tidal} - \vec{U}_{ahf} - \vec{U}_{slip} - \vec{U'}_{alti} \tag{1}$$

In order to yield a geostrophic surface drifter velocity, the processing includes different steps. First, the wind-driven component ($\overrightarrow{U_w} = \vec{U}_{ekman} + \vec{U}_{stokes}$) is estimated and removed (section 5.1). Then the drifter velocities are corrected from the direct effect of the wind on the buoy ($\vec{U}_{slip}$: wind slippage, which is significant only in the case of drogue loss). This is further described in section 5.2. Third the tidal and inertial velocities ($\vec{U}_{tidal}$ and $\vec{U}_{inertial}$), as well as the residual high frequency ageostrophic signal ($\vec{U}_{ahf}$) are removed (see section 5.3). The temporal variability of the drifter geostrophic component is further removed using altimeter data ($\vec{U'}_{alti}$). Finally, the resulting mean synthetic velocities are averaged into 1/8 boxes.

### 5.1 Wind-driven currents modelling

As described in Chapron et al. (2018), under certain conditions (spatial homogeneity of the flow and stationary temporal forcing), the equilibrium between the Coriolis force and the friction force due to wind stress leads to the classical Ekman current formulation (Eq. 2):

$$U_{ekman} = \frac{\sqrt{2}}{\rho_0(f+\omega)\delta} e^{z/\delta}\left[\tau_e^x \cos(\frac{z}{\delta} - \frac{\pi}{4}) - \tau_e^y \sin(\frac{z}{\delta} - \frac{\pi}{4})\right]$$
$$V_{ekman} = \frac{\sqrt{2}}{\rho_0(f+\omega)\delta} e^{z/\delta}\left[\tau_e^x \sin(\frac{z}{\delta} - \frac{\pi}{4}) + \tau_e^y \cos(\frac{z}{\delta} - \frac{\pi}{4})\right] \tag{2}$$

Where $\tau_e$ is the effective wind stress, taking into account the effect of surface-velocity dependency (depending upon oceanic and atmospheric stability correction). $f + \omega$ is the modified vorticity with $2\omega = \partial_x v - \partial_y u$, the local vorticity of the underlying flow ($u,v$). $\delta$ is the upper stress-driven boundary layer, i.e. the Ekman layer depth; it can be expressed as (Pollard et al, 1973):

$$\delta = \gamma^{1/4} w_*/\sqrt{fN} \tag{3}$$

where $\gamma \approx 0.2$, $w_* = \sqrt{\tau/\rho_0}$ , f the Coriolis parameter and N the Brunt-Väisälä frequency.

Following this formulation, the Ekman response to wind stress at the surface is directed at 45° to the right (resp. left) of the wind direction in the northern (resp. southern) hemisphere. Ekman currents further evolve with depth following a logarithmic profile and a further rightward (leftward) spiral. However, near the surface, the classical Ekman spiral model is modified by the influence of surface gravity waves through the inclusion of the Stokes drift. Accordingly, the vertical profile of the quasi-Eulerian current, its magnitude and surface angle, depart from the classical model of Ekman (1905). In particular, Ardhuin et al. (2009) have shown that the surface angle can be much larger than the standard 45° direction. The full surface Stokes drift is estimated to be on the order 0.5-1.3 % of the wind speed and the quasi Eulerian current, $\vec{U}_{ekman} + \vec{U}_{stokes}$ is then of the order 0.6 % of the wind speed, and lies at an average angle between 40° and 70°, to the right



of the wind direction (Northern hemisphere). Near the surface, the surface Stokes drift induced by the waves typically accounts for 2/3 of the total surface wind-induced drift. In this section, the objective is to best estimate the total wind driven component of the current, $(\vec{U_w} = \vec{U}_{ekman} + \vec{U}_{stokes})$ to further remove it from the in-situ drifting buoy velocities. For that purpose, we extend the approach described in Rio et al, 2014a for Ekman current modelling to further include the stokes drift component:

A 2-parameter (β,θ) model (Eq. 4) is used.

$$\vec{u}_w(z) = \beta(z)e^{i\theta(z)}\vec{\tau}^{c(z)} \tag{4}$$

The β and θ parameters are obtained through a least square fit between the wind-driven velocity $\vec{u}_w(z)$ extracted from the drifting buoy measurements and the wind stress values interpolated at the buoy times and positions.

$\vec{u}_w(z)$ is obtained for z=0m by removing the altimeter derived geostrophic velocities from the surface Argo float velocities.

$\vec{u}_w(z)$ is obtained for z=15m by removing the altimeter derived geostrophic velocities from the 15m drogued SVP drifter

velocities and further applying a low pass filter. The filtering cutoff length is taken as the maximum between the 24 hours tidal period and the inertial period. The wind stress along the buoy trajectory is also filtered consistently.

In Rio et al (2014a), c(z) was set to 1 at both level (c(z=0m)=c(z=15m)=1). In this study we determined c(z) at each level so as to maximize the percentage of global variance explained by the model (Eq. 4). We found c(z=0m)=0.6 and c(z=15m)=0.7. For comparison, in the tropical band, Ralph and Niiler (1999) found an optimal value c(z=15m)=0.6.

From Ekman theory (Eq. 2), we expect both β and θ parameters to present regional and seasonal variabilities, in correlation with the varying upper stress-driven boundary layer δ (which depends on latitude and ocean stratification). Indeed, following Eq. (2), in summer, when Ekman depth decreases, β increases and |θ| increases.

In order to take into account these variations, the approach used by (Rio et al, 2011, Rio et al, 2014a) was to fit both parameters by month and by 4° by 4° boxes. In Rio et al (2014a), the wind-driven response at the surface was found to be

located at around 20–40° to the right of the wind direction in the Northern Hemisphere (to the left in the Southern Hemisphere), and the angle then increases to 40–60° at 15m depth. In addition, a clear seasonal cycle was obtained for both parameters and at both depths with larger angles and amplitudes in summer than in winter in good consistency with stronger summer stratification.

In this paper, in order to further resolve the temporal variability of the β and θ parameters, we fit the model from Eq. (4) as a

function of latitude (1° bins) and MLD (5m bins). Weekly MLD values are calculated from the ARMOR3D T/S grids (Guinehut et al, 2012; see section 3) using as criteria the minimum depth between the isotherm layer depth and the isopycnal layer depth (De Boyer Montegut et al, 2004). We checked (not shown) that β and θ values are symmetrical with respect to the equator, so that the parameters for the northern and southern hemisphere are fitted together (β and θ are computed as a function of MLD and |lat|).

The obtained β and θ parameters are displayed in Figure 3. As expected, at a given latitude, both parameters increase with increasing stratification (smaller MLD values). We also obtain a smaller angle response at the surface compared to 15m





depth, and a larger amplitude parameter, in good agreement with an Ekman-like spiral. Finally, as expected from the inverse dependency of the Ekman amplitude response with the Coriolis parameter f (Eq. 2), the β parameter decreases with increasing latitude. We also observe a decrease with latitude of the surface angle parameter. This might be due to an increase

of the Stokes drift amplitude at high latitudes (due to steeper wind waves).  At 15m depth, the angle response is not linear with latitudes, but shows a more complex latitude dependency with a minimum near 30°. The angle is dependent on the Ekman depth (Eq. 2), which depends on many different parameters all varying with latitudes: the wind stress, the Coriolis parameter, and the stratification (through the Brunt-Väisälä frequency).

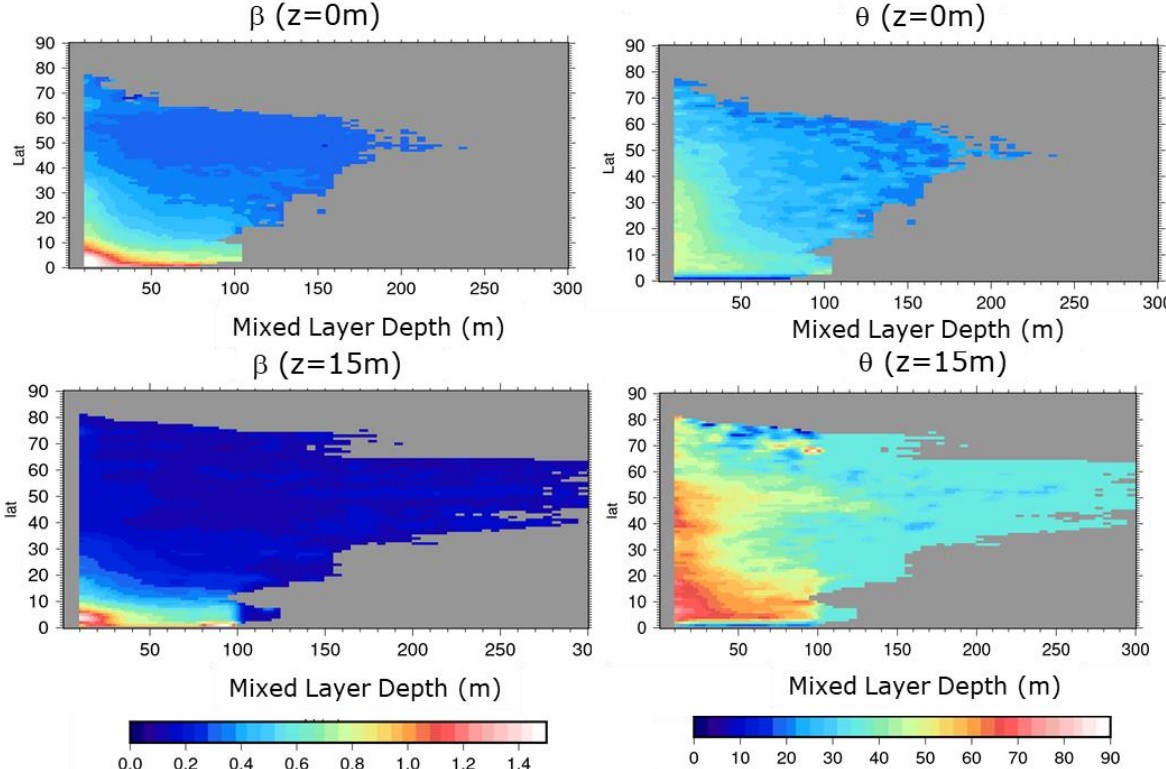

**Figure 3: The β (left) and θ (right)  parameters calculated in this study as a function of Mixed Layer Depth and |latitude|. The top (resp. bottom) plots show the parameters at the surface (resp. 15m depth). Only bins with more than 50 values are shown.**

Another interesting aspect from Figure 3 is that the latitudinal dependency almost vanishes for deep MLD values.

This is expected from theoretical considerations: From Eq. (2) and Eq. (3) we get:

$$\beta(z = 0m) = \frac{\sqrt{2}\sqrt{N}\sqrt{\tau}}{\gamma^{1/4}\sqrt{\rho_0}\sqrt{f}}$$

(5)

$$\beta(z = 15m) = \frac{\sqrt{2}\sqrt{N}\sqrt{\tau}}{\gamma^{1/4}\sqrt{\rho_0}\sqrt{f}} e^{\frac{15\sqrt{fN}}{\gamma^{1/4}w_*}}$$

(6)





In the case of a well mixed upper layer (strong MLD values), the Prandtl number $N/f \approx 1$ so that the β value does not
depend any more on the f values (i.e β becomes independent of latitude).

Table 1 shows the percentage of variance explained by the new surface model, compared to the old one (Rio et al, 2014a).

$$\%VarU = \frac{\sum_i\left(u_w - u_w^{in-situ}\right)^2}{\sum_i\left(u_w^{in-situ}\right)^2} \tag{7}$$

$$\%VarV = \frac{\sum_i\left(v_w - v_w^{in-situ}\right)^2}{\sum_i\left(v_w^{in-situ}\right)^2} \tag{8}$$

The new model is fitted using the entire 2007-2016 period dataset, whereas the old model was based only on the 2007-2014
period, so that results might be biased toward the new solution. To remove this ambiguity, we have repeated the new model
parameter calculation using data from 1997 to December 2014. We find (not shown) that the obtained parameter values are
very similar to the values obtained using the entire period. We then calculate the percentage explained in independent data
(2015-2016). It is much improved for both components of the velocity compared to the previous model from Rio et al
(2014a), especially at latitudes smaller that |5°|. Similar results are obtained for the 15m depth model (Table 2).

| All data (2007-2016) | | | | | | |
|---|---|---|---|---|---|---|
| Surface | All LAT (1113283 data) | | |LAT|>5 (1026446) | | |LAT|<5 (86837) | |
| Model | %VarU | %VarV | %VarU | %VarV | %VarU | %VarV |
| OLD | 31.96 | 19.29 | 34.5 | 20.88 | 22.86 | 11.62 |
| NEW | **35.30** | **21.44** | **37.18** | **22.80** | **28.53** | **14.53** |
| Independent data (2015-2016) – model fitted using 2007-2014 | | | | | | |
| | All LAT (206239 data) | | |LAT|>5 (991460) | | |LAT|<5 (86551) | |
| OLD | 29.04 | 16.62 | 31.53 | 18.12 | 21.08 | 9.33 |
| NEW | **32.64** | **18.61** | **34.12** | **20.11** | **27.90** | **11.33** |

**Table 1: Percentage of variance explained by the surface wind-driven model**



| All data (1993-2016) | | | | | | |
|---|---|---|---|---|---|---|
| 15m | All LAT (14271246 data) | | \|LAT\|>5 (13073431) | | \|LAT\|<5 (1197815) | |
| Model | %VarU | %VarV | %VarU | %VarV | %VarU | %VarV |
| OLD | 12.6 | 10.9 | 12.41 | 12.24 | 9.22 | 8.49 |
| NEW | **15.27** | **12.62** | **14.99** | **12.18** | **13.41** | **11.67** |
| Independent data (2015-2016) – model fitted using 1993-2014 | | | | | | |
| | All LAT (1451989 data) | | \|LAT\|>5 (1346484) | | \|LAT\|<5 (105259) | |
| OLD | 13.0 | 10.2 | 13.82 | 10.86 | 10.8 | 7.45 |
| NEW | **15.67** | **11.35** | **15.33** | **11.67** | **16.37** | **9.93** |

**Table 2: Percentage of variance explained by the 15m depth wind-driven model**

**5.2 Wind slippage**

The SVP-type drifters distributed by the SD-DAC consist of a surface float connected to a subsurface 7-m-long holey sock drogue centered at 15-m depth. Such drifters are designed to minimize the direct action of the wind on the buoy (wind slippage) so that the buoy is advected by the ocean currents at 15m depth (Niiler et al., 1995). When the drogue is lost, the buoy trajectory is due to both the surface currents and the wind slippage.

In order to correct the undrogued drifting buoy velocities for the wind slippage, an updated version of the method described
in Rio et al. (2011) is used. It consists in finding the optimal $\alpha$ coefficient so as to minimize the correlation between the wind stress and a residual velocity $\mathbf{U_r}$ as defined in Eq. (9):

$$\mathbf{U_r}=(\mathbf{U_{buoy}}-\mathbf{U_{ekman}}-\mathbf{U_{stokes}} -\mathbf{U_{geo}})_f - \alpha \, \mathbf{W} \qquad\qquad (9)$$

where $U_{buoy}$ is the drifter velocity, $U_{geo}$ the geostrophic current derived from altimetry and W the wind speed.

In Rio et al. (2011), the vectorial correlation is computed on a 100-day moving window. So only trajectories with more than 100 days can be processed and no wind slippage values are computed for the first/last 50 days of the trajectory, resulting in the loss of a high number of velocity observations.

We therefore upgrade the method in order to calculate the wind slippage correction for the whole drifter's trajectories.

To evaluate the impact of using different correlation window lengths (100 days, 50 days, 30 days, 20 days and 10 days), we use a selection of 21 trajectories longer than 300 days. Results show that below 30 days, the computed alpha coefficient is too noisy, so we choose 30 days as the lower bound of the correlation window $T_L$: 30 days $\leq T_L \leq$ 100 days.

In order to process the first and last $T_L/2$ days of the trajectory portions and also the trajectories shorter than 60 days, two
methods have been investigated:

-     The use of a mean $\alpha$ value, calculated from the $\alpha$ obtained along the drifter trajectory,



- The use of a climatological $\alpha$ value: A mean $\alpha$ is computed for each drifter's trajectories longer than 60 days, and values are then averaged into 4°x4° spatial bins to yield climatological $\alpha$ values.

To validate the different wind slippage corrections, we compare the geostrophic altimetric velocity $U_{geo}$ to ($\mathbf{U_{buoy}}$-$\mathbf{U_{ekman}}$-
$\mathbf{U_{stokes}}$)$_f$  - $\alpha_{\mathbf{xx}}\mathbf{W}$, where the subscript xx is set to:

- Clim for climatological $\alpha$;
- Mean for mean $\alpha$ value;
- 0 when no wind slippage correction is applied ($\alpha_0 = 0$);
- N for the nominal $\alpha$ ($T_L = 100$ days).

We compute Root Mean Square Differences for both components of the velocity (RMSDU and RMSDV).

The first, expected, result, is that applying a wind slippage correction always yields better results than not applying any correction: RMSDU (RMSDV) is reduced from 13% to 7% (from 10% to 6.5% ). The second result is that using a mean $\alpha$ along the drifter trajectory yields better result than using the climatological $\alpha$ value. The RMSDU and RMSDV are reduced by 2.2%.

Consequently, we finally apply the following procedure to remove the wind slippage from the drifting buoy velocities: the first and last $T_L/2$ days of the trajectories longer than 60 days are completed using $\alpha_{MEAN}$. Once all drifter trajectories longer than 60 days have been processed, $\alpha_{CLIM}$ is used to correct the trajectories shorter than 60 days. Note that a higher error is associated with velocity estimate at the beginning and end of the trajectory since the wind slippage correction is less accurate. It corresponds to 21% of the total amount of drifters for the mean $\alpha$ and 1.5% for the climatological $\alpha$.

**5.3 Synthetic mean velocities calculation**

The synthetic mean velocities are then calculated following Eq. (1): The 15m drogued SVP velocities are corrected from the wind-driven component using the empirical model from Eq. (4) and the $\beta(z=15m)$, $\theta(z=15m)$, $c(z=15m)$ parameters. Undrogued SVP-type drifters are first corrected from the wind slippage (section 0.2) and then from the wind-driven component using the empirical model from Eq. (4) and the $\beta(z=0m)$, $\theta(z=0m)$, $c(z=0m)$ parameters. In order to remove the
tidal and inertial components from the drogued and undrogued SVP drifters, a low pass filter is then applied along the drifter trajectory. In past work (Rio et al, 2004, 2012, 2014a) a unique 3-day filtering length is used. Here we refine the filtering length by considering for each drifter trajectory  the lowest latitude sampled (absolute value) and filter the trajectory using a cutting period $P_c = max(P_i, 24 hours)$ where $P_i$ is the inertial oscillation period at the lowest latitude, so as to be sure to filter the main tidal currents, the inertial oscillations and any residual high frequency ageostrophic signal.

The Argo floats' surface velocities are corrected from the surface wind-driven model. Argo floats surface velocities being sampled only every 10 days, no further low-pass filtering can be applied on the Argo velocity dataset. Also, as discussed in



Rio et al. (2011), no wind slippage correction is applied to Argo floats since, thanks to their design, they are less impacted by wind slippage than undrogued SVP.

Temporal variability is then removed by interpolating altimetric geostrophic velocity anomalies along the drifter trajectories
and subtracting them from the in-situ geostrophic velocities. The obtained mean geostrophic velocities are further averaged into 1/8° boxes (instead of ¼° boxes in Rio et al (2014a)) and displayed on Figure 4. Associated errors are obtained as described in Rio et al. (2011).

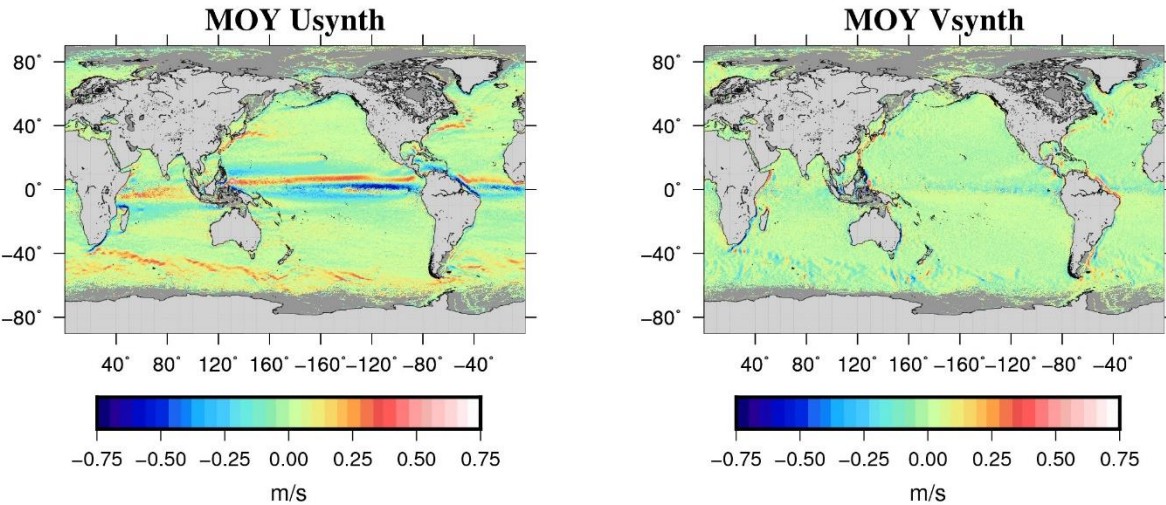

**Figure 4: Mean zonal (left) and meridional (right) synthetic velocities**

**6 Dynamic Heights data processing**

The method used to calculate the synthetic MDT ($MDT_{synth}$) is the same as the one fully described in Rio et al. (2011) and also used in Rio et al. (2014). We use the temperature (T) and salinity (S) profiles from the CORA database (section 0) to calculate instantaneous dynamic heights $h(t,r)_{/Pmax}$ relative to a maximum profile depth $P_{max}$. Following the same philosophy as in the computation of synthetic velocities, the temporal variability is subtracted from $h(t,r)_{/Pmax}$ using altimetric anomalies
(SLA) referenced over the 1993-2012 time period to end up with a mean field $\langle h \rangle(r)_{/Pmax}$ also referenced over 1993-2012 time period. The resulting mean dynamic heights $\langle h \rangle$ represents only the baroclinic processes from the surface down to $P_{max}$. We need to add the mean contribution of deep baroclinic and barotropic processes. The missing quantity is evaluated as the difference between a dynamic height climatology referenced at $P_{max}$ ($h_{clim/Pmax}$) and the CNES-CLS18 MDT first guess (Eq. 10). To avoid any large-scale differences and time period discrepancy between $\langle h \rangle(r)_{/Pmax}$ and the climatology $h_{clim/Pmax}$, a
specific climatology is computed using the same observations from CORA while a WOA climatology was used by Rio et al. (2011).



$$MDT_{synth} = <h>(r)_{/Pmax} - h_{clim/Pmax} + First\ Guess \tag{10}$$

The calculation of the CNES-CLS18 MDT is based on a remove-restore technique where the MDT first guess is first removed and then added back to the optimally estimated field. Thus, the useful quantity in Eq. (10) is the difference between $<h>_{/Pmax}$ and its climatology $h_{clim/Pmax}$, i.e. the small scales of the mean dynamic heights referenced at $P_{max}$. Indeed, the aim of

the objective analysis is to optimally add scales smaller than those resolved by the MDT first guess. Most of the T/S profiles go down to 2000 m, thus Figure 5 illustrates the difference between $<h>_{/Pmax}$ and its climatology $h_{clim/Pmax}$ for $P_{max}= 2000m$. The signal is high for instance in the Antarctic Circumpolar Current with anomalies higher than 10 cm and in the equatorial currents system area mainly in the western Pacific with anomalies around ± 7 cm. We thus expect that these in-situ data will bring useful information in these areas.

Finally, the synthetic mean heights are averaged in ¼° boxes. Note that, unlike the drifters, we do not have enough observations of height to average in 1/8° boxes. The associated errors are computed as described in Rio et al. (2011). There are dominated by oceanic variability patterns, with errors up to 10 cm in high variability areas (western boundary current, Antarctic Circumpolar Current - ACC) and less than 2 cm far from these areas.

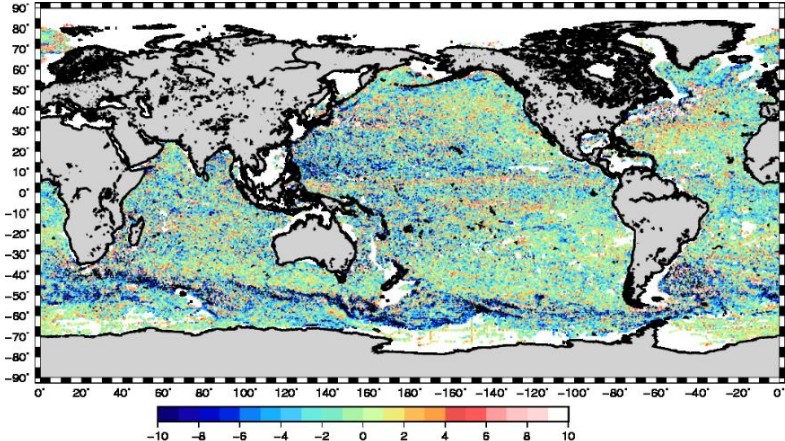


**Figure 5: <h>(r)/2000m - h_clim/2000m (cm)**

**7 High resolution MDT inversion**

The synthetic mean geostrophic velocities and mean heights calculated in the previous sections are then used to improve the GOCE based first guess using a multivariate objective analysis as in Rio and Hernandez (2004) and Rio et al (2005, 2007,

2011, 2014a). The method relies on the prescription of the apriori MDT variance and the apriori spatial zonal and meridional spatial correlation scales of the estimated field. We use the same apriori statistics as in Rio et al (2014a). The obtained CNES-CLS18 MDT is displayed on Figure 6.




Estimates of the mean geostrophic velocities are obtained as output of the multivariate objective analysis. In the equatorial band, only the synthetic mean velocities are used for the inversion in order to circumvent the issue due to the failure of the

geostrophic approximation. For the same reason, only the synthetic height is used to estimate MDT in the equatorial band. The right plot of Figure 6 shows the norm of the obtained mean currents. All currents are significantly enhanced compared to the speed of the mean currents from the GOCE based first guess (not shown).

a)                                                                    b)

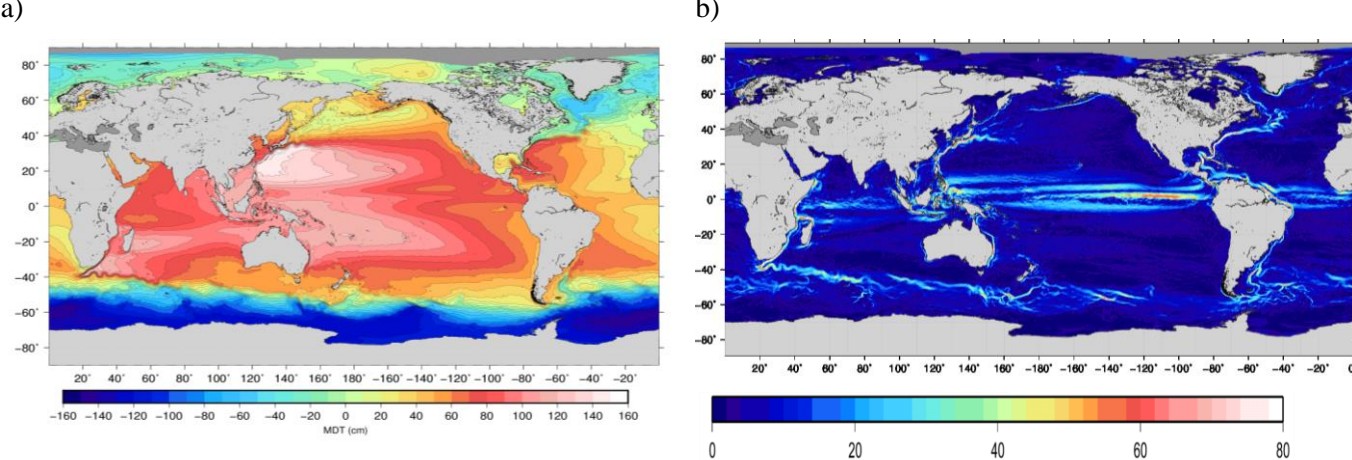

**Figure 6: (a) The CNES-CLS18 Mean Dynamic Topography (cm) and (b) intensity of the associated mean geostrophic velocities**

A spectral analysis tool is used to investigate the spatial scales of the different MDT solutions : first guess, CNES-CLS13

MDT, new CNES-CLS18 MDT and Glorys12 numerical model MDT (a 1/12° numerical model from Mercator-Ocean (Artena et al., 2019)). An example is given in Figure 7 for a 10° box in the Agulhas Return Current area. We clearly see the increased energy at short scales contained in the new CNES-CLS18 MDT (in green) compared to either the GOCE based first guess (in purple) or the previous CNES-CLS13 MDT (in blue). The energy level is in good agreement with the GLORYS12 energy level (in red) until 50 km wavelength (25 km resolution). At shorter scales a flat response is obtained.

This means that the shortest scales (between 12.5 km and 25 km) of the CNES-CLS18 MDT might be contaminated by noise.



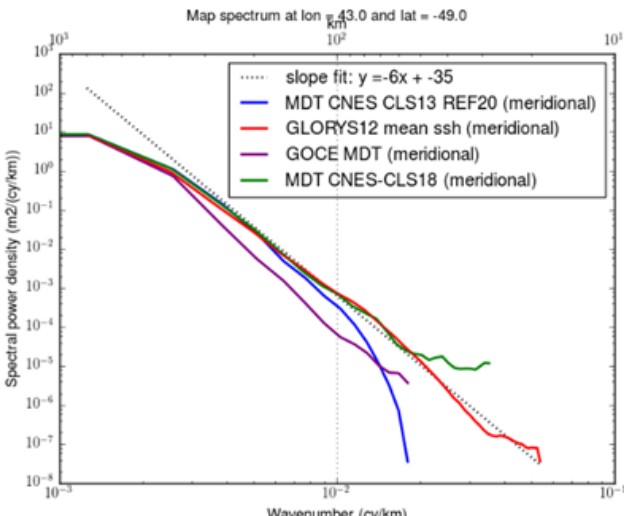

**Figure 7: Spectral Power Density obtained for different MDT solutions**

## 8 Validation

The validation of the obtained MDT-CNES-CLS18 solution uses three different approaches:

1- a qualitative validation in specific areas (Japan Sea, Gulf of Maine, Chilean coast, Agulhas current) to highlight the enhancement of coastal currents and western boundary currents in those areas compared to the previous CNES-CLS13 solution;

2- a quantitative regional validation around Australia and in the Drake Passage;

3- a global, quantitative validation by comparison to an independent dataset of drifting buoy velocities: We use the 15m drogued SVP drifter velocities available for the year 2017 (not used in the calculation of the synthetic velocities).

### 8.1 Qualitative validation

### 8.1.1 The Japan Sea

The Japan Sea is characterized by a number of thin coastal currents as depicted for instance by Figure 1 from Lee et al. (2009). Most of these currents are poorly resolved (or not resolved at all) by the 1/4° resolution CNES-CLS13 MDT (Figure 8, right). In particular, the Liman cold current (LC), the Soya Current (SC), the Nearshore branch (NB) of Tsushima Warm current and the East Korean Warm Current (EKWC) are nicely resolved in the new CNES-CLS18 MDT. None of these are resolved in the GOCE based first guess, so this short scale information is only due to the inclusion of the in-situ information.

The Liman cold current is not at all resolved in the previous CNES-CLS13 solution.





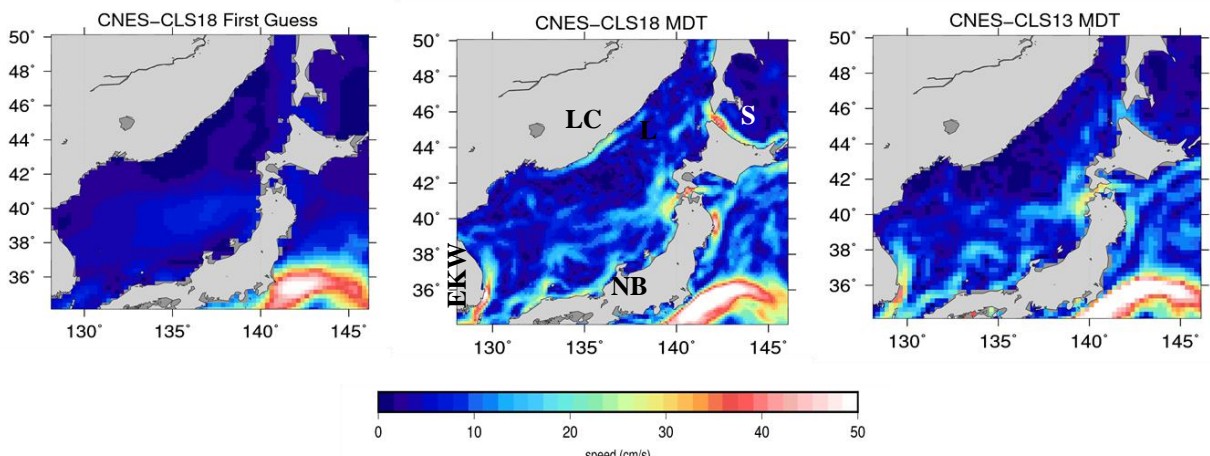

**Figure 8: Mean geostrophic velocities in the Japan Sea in the CNES-CLS18 GOCE based First Guess (left), in the CNES-CLS18 MDT (middle) and in the CNES-CLS13 MDT (right). Abbreviations for the main currents are as follows: LC: Liman Current, EKWC: East Korean Warm Current, NB: Nearshore Branch of the Tsushima Warm current and SC: Soya current.**

### 8.1.2 Gulf of Maine

The freshwater mass enters the eastern Gulf of Maine from south of Nova Scotia shelf and follows a general counterclockwise circulation with a significant contribution to the Maine Coastal Current (MCC). MCC generally flows westward along the coast between 43.5N and 44.5N and creates signals in the MDT: the MDT increases toward the coast. As shown by Feng et al. (2018), this coastal current is not resolved by the MDT CNES-CLS13 while it is resolved by an independently developed $MDT_{RU}$ from a regional ROMS numerical model by Rutgers University that assimilates various in-situ measurements such as T/S profiles, HF-radar and drifters (Levin et al., 2018).

An update of this study is done here including MDT CNES-CLS18 to verify if the new MDT resolves this coastal current pattern (Figure 9, Figure 10). Figure 9 shows a comparison, at six buoy sites, of the mean geostrophic currents inferred from MDT CNES-CLS13 and MDT CNES-CLS18 with ADCP mean upper ocean currents (averaged from the surface down to 50 m and over the whole buoy operated time period: roughly July 2001 to July 2019). Specifically, at the offshore-most buoy N deployed in the eastern side of the northeast channel (NEC), both MDTs agree very well with the buoy mean in both magnitude and direction. Along the Maine coast at buoys I , E and B, the improvement of MDT CNES-CLS18 is obvious although disagreement still exists to some degree at buoys I and B in the western coast. At buoy L, MDT CNES-CLS18 looks reasonable in magnitude and direction, but not consistent with buoy measured one, most likely because at this location ADCP operates over a shorter time period (7/2001 - 8/2008) leading to larger uncertainty. At buoy M deployed at the deeper Jordan Basin (JB), MDTs and the buoy mean are not in agreement, may be because at this location the ratio 'temporal variability' over 'mean signal' is high and thus the comparison of current averaged over different time period is not accurate.





365 Even if the mean currents from ADCP represent a mean over a different time period than the MDTs referenced over 1993-2012: these comparisons show that the mean MCC estimated by MDT CNES-CLS18 is significantly improved compare with the MDT CNES-CLS13. Indeed, Figures 10 shows that the MCC has a signature in the MDT CNES-CLS18 that increase toward the coast while it is not the case in the MDT CNES-CLS13.

As highlighted on Figure 11 and Figure 12, this improvement is due to the better representation of this current both in the first guess and in the mean synthetic velocities used in the computation. First, thanks to the improvement of the processing,

370 the first guess is improved close to the coast and the new first guess MDT increase toward the coast, which is not the case for the MDT CNES-CLS13 first guess (Figure 11). Second, the number of available drifters, the improved processing of this dataset, and their averaging performed in 1/8 boxes instead of ¼ boxes (Figure 12), are all key elements leading to the improvement of the CNES-CLS18 in this specific area.

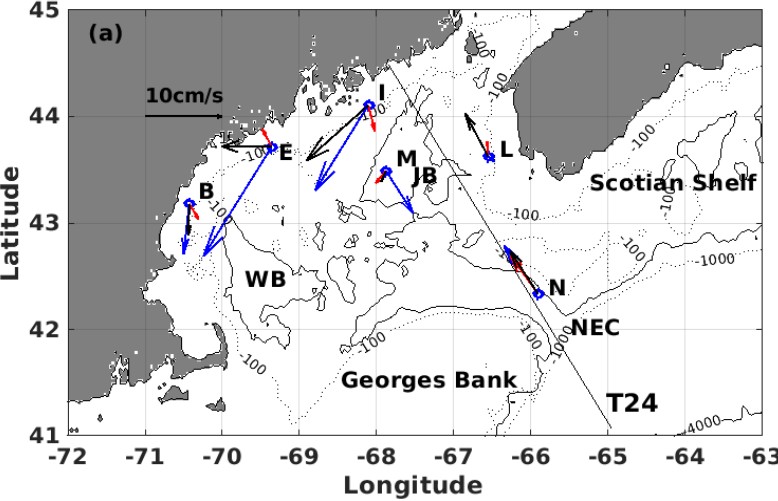

375 **Figure 9: map of Gulf of Maine with the mean upper ocean (50m averaged) current vectors from in situ measurements at 6 buoy sites (N, L, M, I, E and B) in blue, and the mean geostrophic velocity vectors inferred from MDT_CNES_CLS13 in red and MDT_CNES_CLS18 in black. One Jason altimeter track 24 (T24) is also shown. (Abbreviations: JB: Jordan Basin; WB: Wilkinson Basin; NEC: Northeast Channel).**

Figure 10: (a) MDT CNES-CLS18 and (b) MDT CNES-CLS 13 in the Gulf of Maine

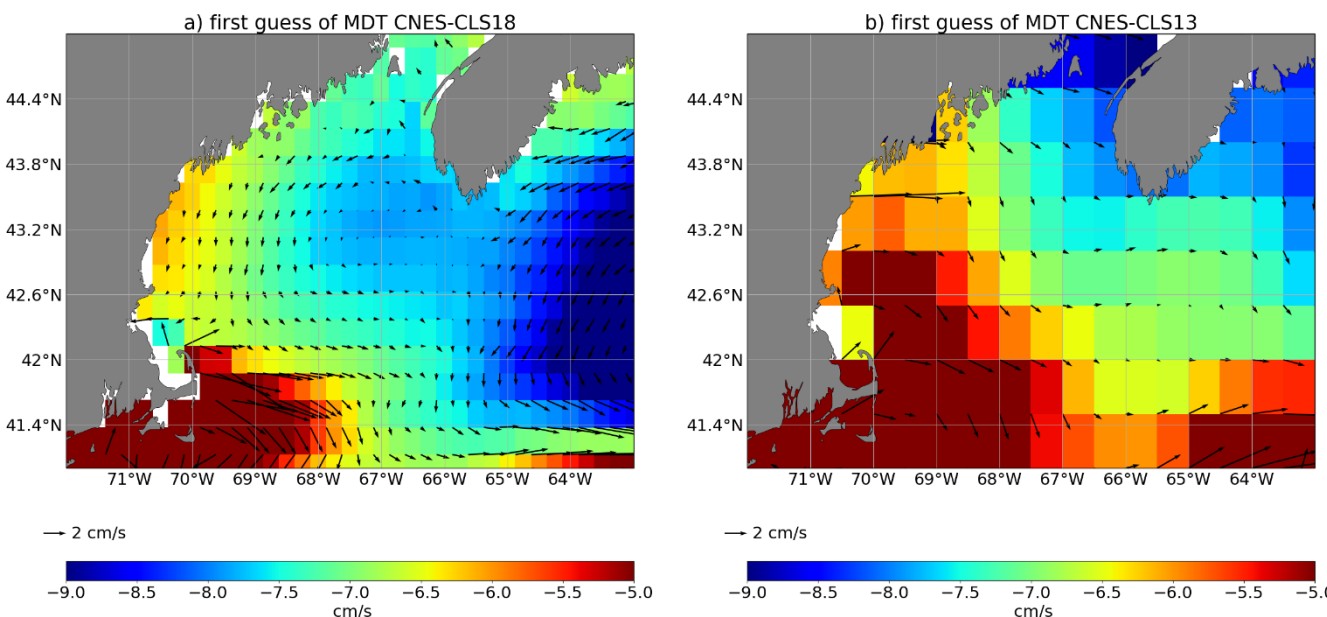

Figure 11: First guess of the MDTs (a) CNES-CLS18 and (b) CNES-CLS13 in the Gulf of Maine





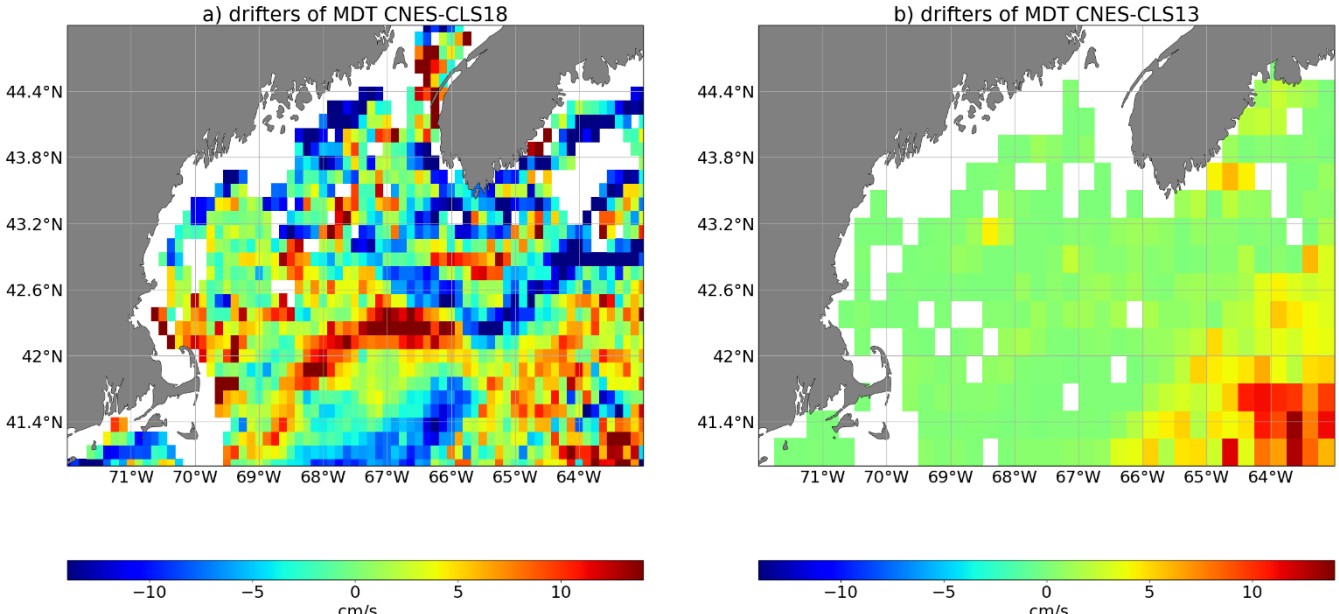

**Figure 12: Zonal component of the mean synthetic velocities from drifters used in the computation of the (a) MDT CNES-CLS18 and (b) MDT CNES-CLS13**

### 8.1.3 Chilean coast

Over the South Chilean coast, the eastward South Pacific Current (SPC) bifurcates to become the Humboldt Current (HC) that flows northward and the Cape Horn Current (CHC) that flows southward. The circulation in this area is fully described by Strub et al. (2019). The HC and CHC are much better resolve in the MDT CNES-CLS18 compared with the previous MDT CNES-CLS13. The circulation resolved by the MDT CNES-CLS18 is thus in better agreement with what is expected and with numerical models (Figure 13). As in the Gulf of Maine, this improvement of the MDT CNES-CLS18 is due to the new drifters dataset and the improved processing of the first guess. The mean circulation along the Chilean coast is difficult to be accurately resolved by the geodetic MDT due to the influence of the Andes mountain and the very rugged coastline. The improved processing of the first guess helps to overcome this issue.

However, offshore of Chiloe Island, the MDT CNES-CLS18 shows an eddy that does not appear realistic. In this area, between the HC and the CHC, one or two observations that have an unrealistic, strong velocity may introduce some noise. We expect that this should be found in other areas and can explain the noise underlined by the spectral analysis at the smaller scales (see section 7).




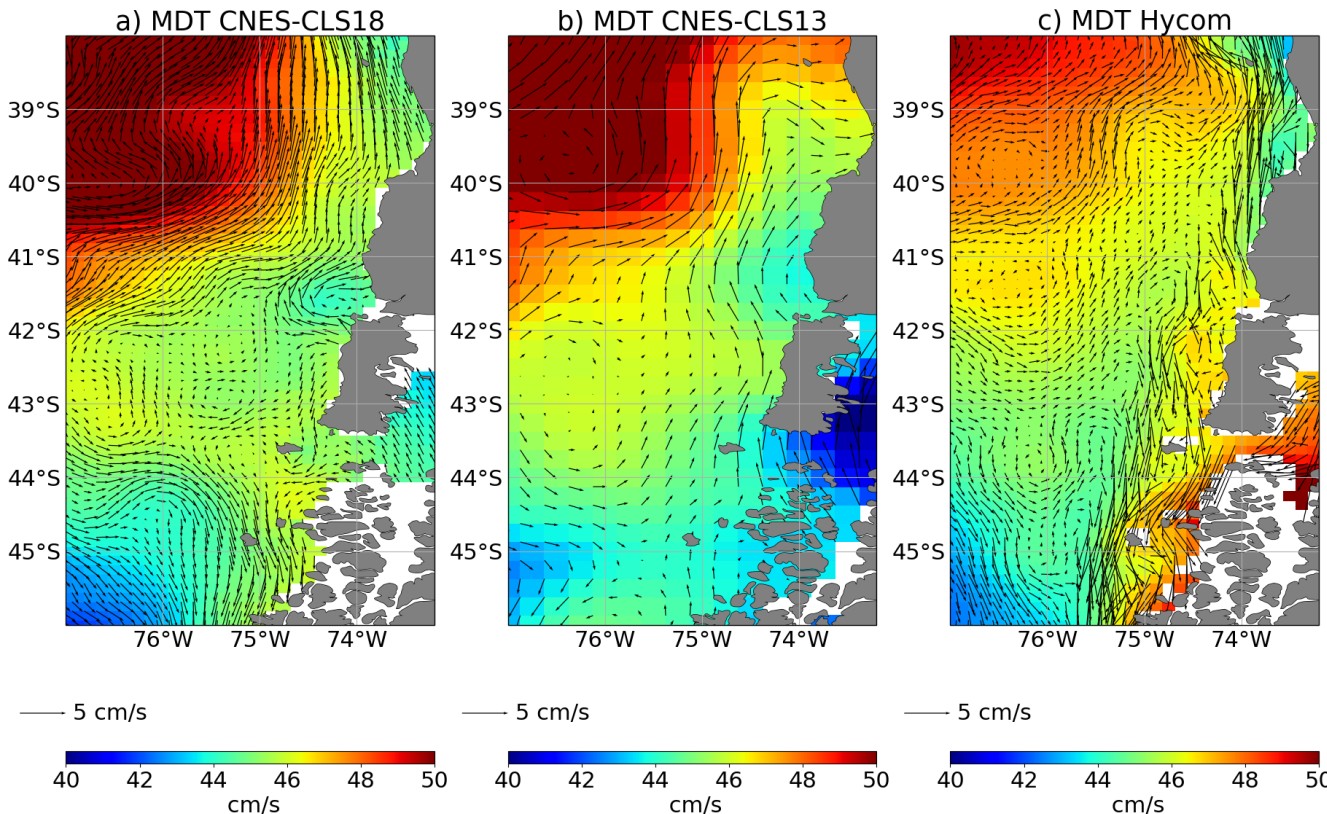

**Figure 13: MDT and associated mean geostrophic current along the south Chilean coast from the MDT CNES-CLS13, MDT CNES-CLS18 and the numerical model HYCOM interpolated on the MDT CNES-CLS18 grid (1/8°)**

### 8.1.4 Agulhas current

As first demonstrated by Chapron et al. (2005), highly valuable information on ocean surface currents can be retrieved by analysing the Doppler shift measured between the signal emitted by a Synthetic Aperture Radar (SAR) antenna and the signal backscattered by the sea surface.

Within the ESA Globcurrent project, mean velocities were calculated in the Agulhas current area from the ENVISAT SAR Doppler velocities. The ENVISAT data span the period from 2007 to 2012. The main limitation of this technique is that only the radial component of the velocity is retrieved. To deal with this limitation, the total current velocities are estimated by using the direction given by the altimeter-derived geostrophic currents, whose norm is corrected using radial component measured by the SAR from either the ascending or descending passes. The corrected norm using the ascending ($\overset{*}{V_a}$) or the descending ($\overset{*}{V_d}$) passes are given by Eq. (11) and (12) respectively:

$$\overset{*}{V_a} = \frac{V_a^{SAR}}{cos(\beta_a)}$$





$$\overset{*}{V_d} = \frac{V_d^{SAR}}{cos(\beta_d)} \tag{11}$$

$$\tag{12}$$

Where $V_a^{SAR}$ and $V_d^{SAR}$ are the SAR-derived radial velocities in ascending and descending passes and $\beta_a$ and $\beta_d$ are the angles between the SAR range direction and the altimeter-derived current direction for ascending and descending passes.

The obtained mean velocities corresponding to the 1993-2012 time period are displayed on Figure 14 (right plot). Velocities

along the African coast are in good agreement with the mean velocities obtained from drifters (middle plot), with maximum values reaching up to 2 m/s in the core of the current. This is around twice the intensity obtained from the GOCE derived MDT First guess (left plot).

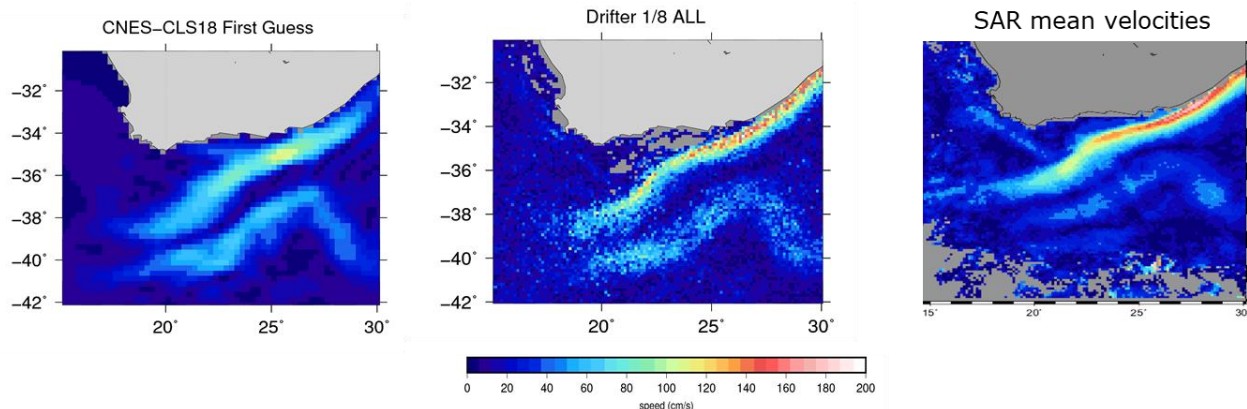

**Figure 14: Mean velocities in the Agulhas Current area from (left) the GOCE-only derived MDT First Guess, (middle) drifting buoy velocities and (right) SAR derived mean velocities**

By using the drifter velocities to improve the GOCE derived first guess, much higher velocity speeds are obtained (right plot of Figure 15). Maximum values reach 1.6 m/s, which is around 30% more than the mean velocities obtained in the previous CNES-CLS13 MDT (maximum velocities are around 1.2 m/s). This is due to the higher resolution of the solution (1/8

instead of ¼), an improved processing of the drifters (less filtering is applied), and the removal of a minimum number of data threshold which was applied in the CNES-CLS13 MDT calculation, removing a number of very coastal in-situ data for the inversion.

In the return current, a discontinuity is observed in the CNES-CLS18 MDT in the speed intensity east and west of around 24 E longitude. This is in qualitative agreement with both the drifter data and the SAR measurements (Figure 14). Instead, due

to coarser resolution, the Agulhas return current intensity is very much continuous along its path both in the CNES-CLS13 MDT (left plot of Figure 15) and the GOCE MDT First guess (left plot of Figure 14).



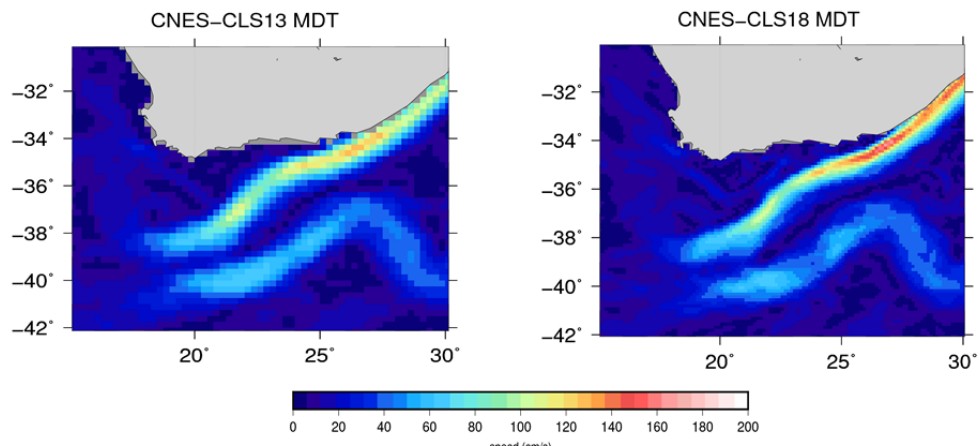

**Figure 15: Mean geostrophic velocities in the Agulhas Current area derived from the CNES-CLS13 MDT (left) and the CNES-CLS18 MDT (right)**

**8.2 Quantitative regional validation**

**8.2.1 Australia**

A number of coastal currents flow around Australia, in particular the Leuwin current (along the West coast), the East Australian Current, and the Hiri current (along the Northern tip of Australia). None of these currents are resolved in the GOCE based first guess (left plot on Figure 16) while they are nicely resolved in the CNES-CLS18 MDT (middle plot of

Figure 16). The Leuwin current was hardly resolved in the previous CNES-CLS13 MDT (right plot of Figure 16).

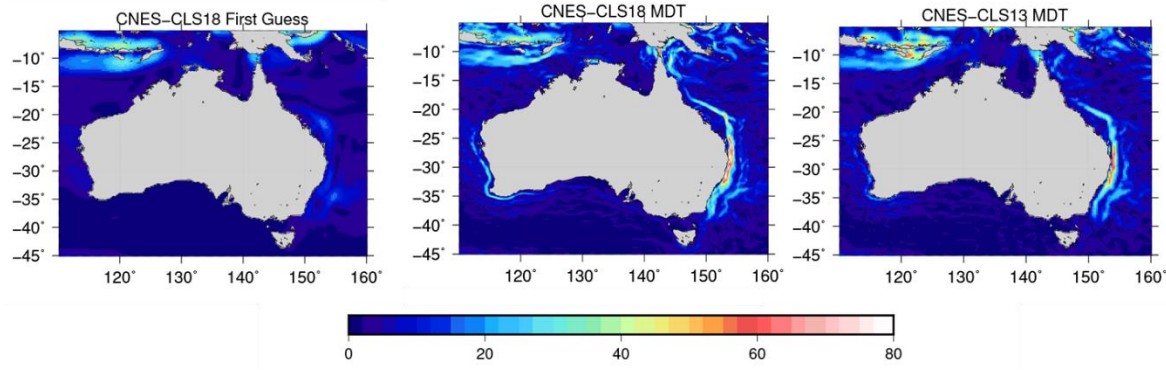

**Figure 16: Mean geostrophic velocities around Australia in the CNES-CLS18 GOCE based First Guess (left), in the CNES-CLS18 MDT (middle) and in the CNES-CLS13 MDT (right)**

Cancet et al. (2019) validate the Copernicus-Globcurrent product that is distributed by CMEMS (https://resources.marine.copernicus.eu/documents/QUID/CMEMS-MOB-QUID-015-004.pdf). They compare the version of



the Copernicus-Globcurrent currents that use the MDT CNES-CLS13 (Rio et al., 2014) with 3 ADCP time series located within the East Australian Current close to the coast. Figure 17 is an update of the results from Cancet et al. (2019) at the station SYD100 (lon = 151.3821°E ; lat = 33.9438°S), including a comparison with a version of Copernicus-Globcurrent

based on the MDT CNES-CLS18 (only the MDT field is changed). When the MDT CNES-CLS13 is used, the current does not seem to have a predominant direction. On the contrary, when the MDT CNES-CLS18 is used, the ocean current rose is in good agreement with the ADCP data, with the main current flowing with a 250° angle.

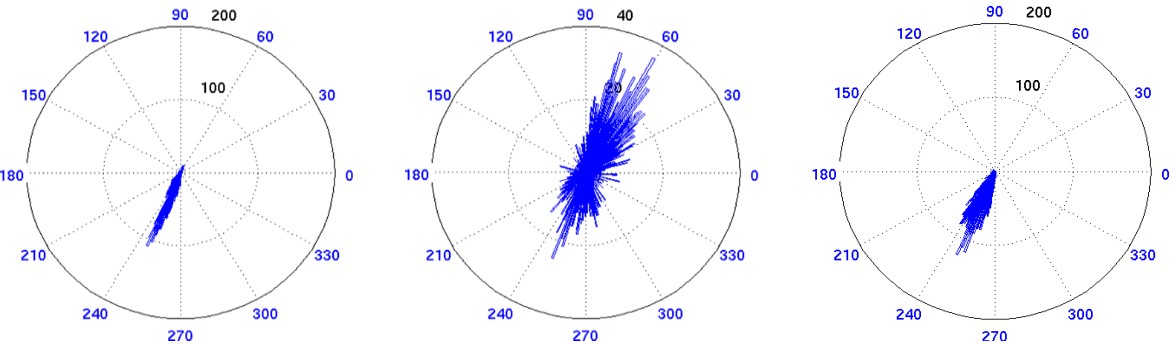

**Figure 17: Ocean current roses of the (left) 30-day low-pass filtered ADCP observations, (middle) Copernicus-Globcurrent based**
**on MDT CNES-CLS13 and (right) Copernicus-Globcurrent based on MDT CNES-CLS18. The numbers in blue indicate the angle of the ocean current direction (0° is eastward, 90° is northward). The numbers in black and the concentric circles give the number of occurrences in the observations for each angle. Note that for the middle plot, the scale is different regarding the occurences.**

### 8.2.2 Drake Passage

The persistent westerly winds over the Southern Ocean drive the Antarctic Circumpolar Current (ACC) flow around the
Antarctic continent. The ACC flow is organized in three oceanic frontal systems, which correspond to water mass boundaries as well as deep-reaching jets of eastward flow: the Subantarctic Front (SAF), the Polar Front (PF), and the Southern ACC Front (SACCF) (Figure 18). The Drake Passage, located between the South American and Antarctic continents, is the narrowest stretch of water (about 850 km wide), separating Antarctica from the northern continents, thus it forms the most practical location to monitor the ACC (Figure 18). In the context of the DRAKE project (Provost et al., 2011)
hydrographic data and current meter time series were collected during 3 years (2006–2009) in the upper 3000 m of the water column below the ground track 104 of Jason altimetry satellite (Figure 18, black section and white dots). Combining the DRAKE in situ mooring and satellite altimetry velocities Koenig et al. (2014) build a Look-up-Table (LUT) to compute the ACC transport. Koenig et al., (2014) adjust the mean cross-track surface velocity at each mooring location using an iterative error/correction scheme with the CNES-CLS13 mean surface geostrophic velocities as a first guess. The three mean surface
geostrophic velocities from the CNES-CLS09 (Rio et al., 2011), CNES-CLS13 (Rio et al., 2014) and CNES-CLS18 MDTs are compared to the mean surface geostrophic velocities from Koenig et al. (2014) in Figure 19. The new CNES-CLS18 MDT provides mean surface geostrophic velocities consistent with those from Koenig et al. (2014). Both present a strong SAF (located at 56.2°S) and SACCF-S (60.5°S). They show two recirculation cells: negative velocities around 57°S and at





59.7°S as identified in Ferrari et al. (2012 and 2014) with the DRAKE current-meter data. This is an improvement as the
recirculation at 57°S is not captured in the velocities derived from the CNES-CLS09 and CNES-CLS13 MDTs. The mean
velocities derived from the CNES-CLS18 MDT further indicate a strong current close to the coast reaching 35 cm/s at 55°S.
This feature is observed in all the L-ADCP sections (Renault et al., 2011) and in GLORYS12 reanalysis (Artana et al.,
2019). We propose an adjustment of the mean geostrophic surface velocities from Koenig et al. (2014) (LUT- updated, red
dashed line) close to south American continent to account for this coastal current associated with a northern branch of the
SAF. This coastal current carries slightly less than 2 Sv, increasing the mean Koenig at al. (2014) ACC transport to 143 Sv.
The MDTCNES-CLS18 is also shown to perform better than the MDT13 in the SAF at 41°S (Malvinas Current),
downstream of Drake Passage, in Artana et al. (2019).

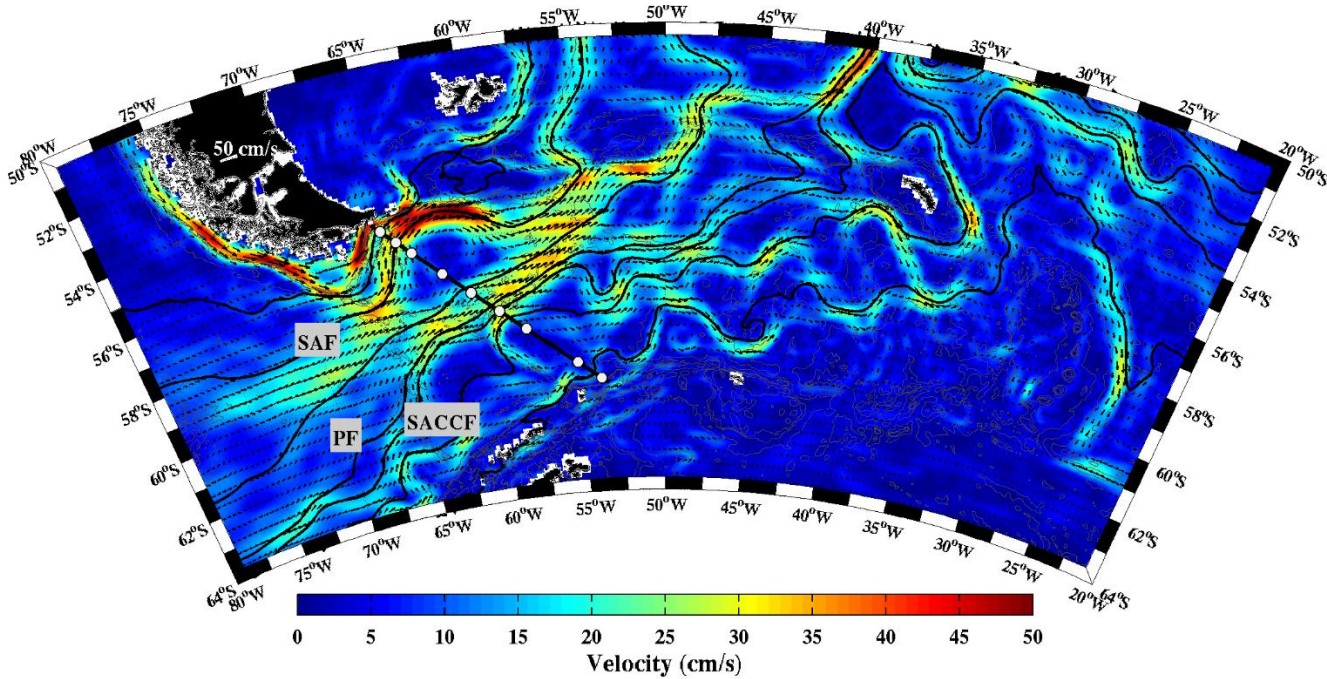

**Figure 18: Mean surface velocity magnitude (in cm/s) and mean surface velocity vectors from MDT CNES-CLS18. Black lines are**
**isolines of mean sea surface height corresponding to the main Antarctic Circumpolar Current (ACC) fronts and their branches:**
**SAF-N (23 cm), SAF-M ( 0 cm), PF-N (43 cm), PF-M (-70 cm), PF-S (-79 cm), SACCF-N (-93 cm)and SACCF-S (-114cm). White**
**dots indicate the position of the DRAKE moorings.**



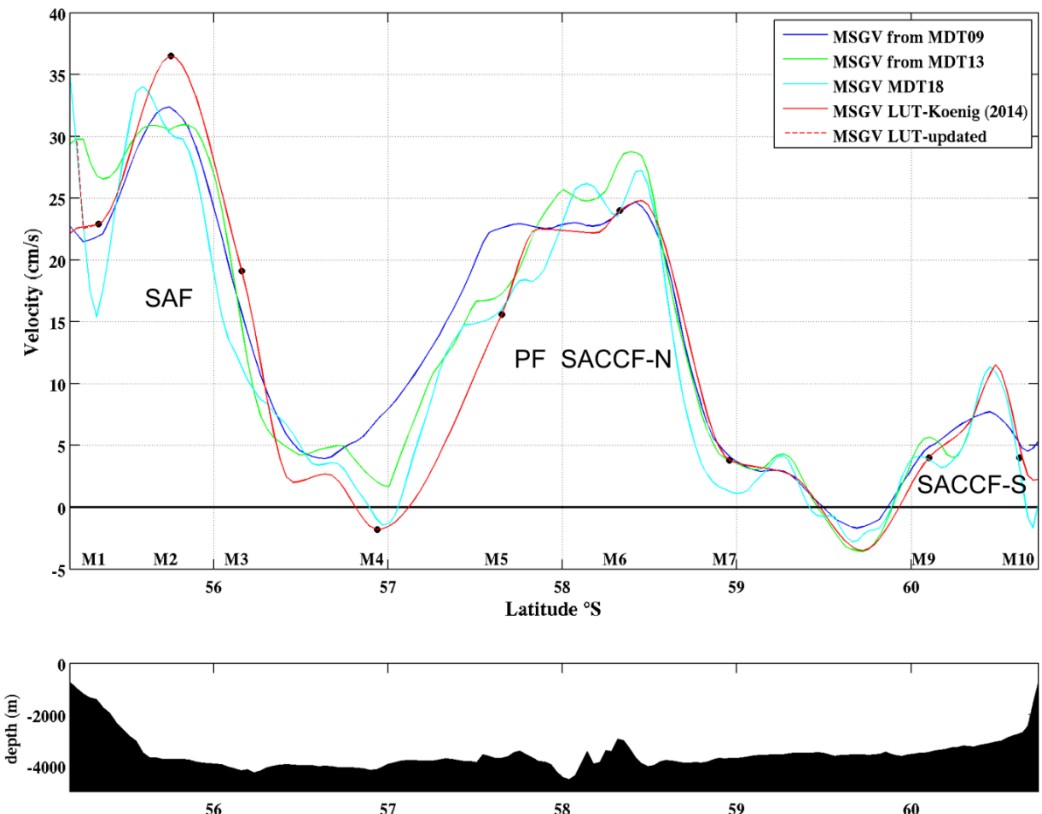

**Figure 19: Mean surface geostrophic velocities (MSGV) derived from MDTCNES-CLS09 (blue), from MDTCNES-CLS13 (green),**
**MDTCNES-CLS18 (cyan) and LUT (red) along Jason track #104. Location of the Drake moorings are indicated M1 to M10.**
**Below bathymetry along the section.**

## 8.3 Quantitative validation through comparison to independent drifter velocities

The number of available independent 15m drogued drifters for year 2017 is shown in Figure 20. The world ocean is rather
well sampled, apart from the Indonesian throughflow, the Arctic Ocean, and the Southern Ocean (latitudes poleward of 60°
S).



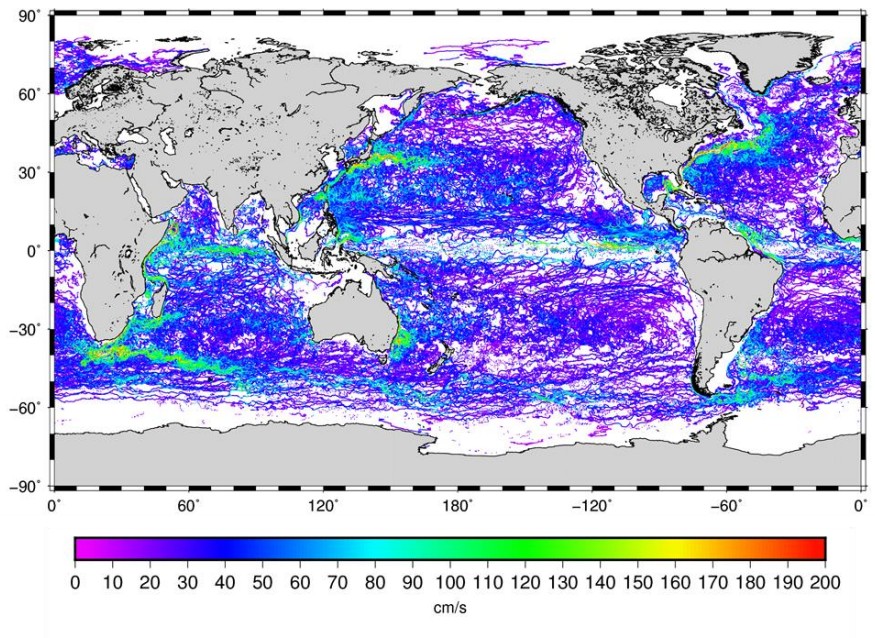

**Figure 20: Trajectories of the 15m drogued SVP drifters available over year 2017. Colors give the speed velocities deduced along the drifter trajectories.**

Absolute dynamic topography values are calculated by adding gridded Sea Level Anomalies from DUACS-2018 (section 3) to the new CNES-CLS18 MDT. Corresponding geostrophic velocities were then derived and interpolated along the drifter trajectories. Root Mean Square differences (RMSD) between the obtained geostrophic velocities and the drifter derived geostrophic velocities for year 2017 are then calculated both for the zonal and meridional components. The same comparison is performed using the CNES-CLS18 first guess, the previous CNES-CLS13 MDT and its first guess.

The RMSD of the independent velocity dataset are then calculated as a function of distance to the coast (Figure 21). Everywhere, the RMSD is reduced with the MDT CNES-CLS18 and especially in coastal areas (distances from the coast less than 100 km). Part of this improvement is due to first guess improvement, since the RMSD is also reduced when the CNES-CLS18 first guess is used rather than the CNES-CLS13 first guess. This was already pointed out in the Gulf of Maine and along the Chilean coast. As described in section 8.1, this improvement is also due to in-situ data, especially to processed
drifters.





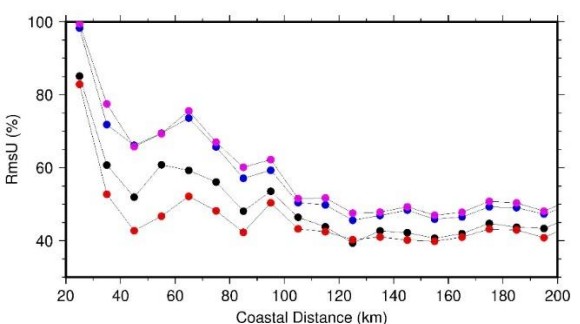
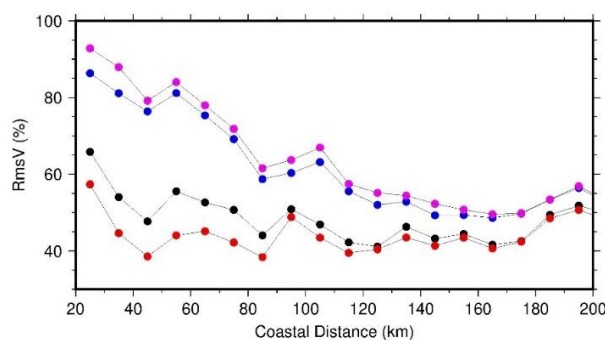

**Figure 21: Zonal (left) and Meridional (right) RMS differences as a function of coastal distance between the independent drifting buoy velocities and the altimeter geostrophic velocities obtained using different MDT solutions: Pink the CNES-CLS13 MDT first guess; Blue the CNES-CLS18 MDT First Guess; Black the CNES-CLS13 MDT and Red the CNES-CLS18 MDT**

**9 Conclusions**

The new CNES-CLS18 MDT is presented, focussing on the innovative elements in the processing and datasets used compared to the previous CNES-CLS13 MDT solution, and on validation results at global, regional and coastal scales. Among the main improvements, a more recent GOCE geoid model is used and the calculation of the first guess is improved. More in-situ hydrological profiles and drifting buoy data are used and a new wind slippage correction and wind driven

model are computed to better extract the geostrophic current from the drifters. Grid resolution is enhanced from 1/4° for earlier solutions to 1/8°. The effective resolution is also improved as shown by spectral analysis. Thus, this new MDT better retrieves the strongest ocean currents and shows huge improvement in coastal areas, as highlighted by a number of specific coastal and regional examples. Indeed, narrow and coastal currents are now resolved, which are not in the previous solutions. The continuous improvement of MDT accuracy and resolution is necessary, all the more in the context of the upcoming

swath altimetry missions (e.g SWOT, whose launch is planned in 2021) that will be able to retrieve smaller spatial scale of the Sea Surface Height. Moreover, as underline by Hamon et al. (2019), an accurate MDT and its error are important to correctly assimilate altimetric observations in numerical model. Further processing improvements might be considered. First, the computation of the first guess could be improved by using new method such as the one described by Siegismund et al. (2020). In particular, this could help to better resolve MDT along the coast. Second, we should address the residual noise

issue highlighted in some areas at scales shorter than 25 km. This might be done by refining the errors prescribed on the drifter derived velocities. We need also to better understand and quantify the residual temporal variability and how the spatio-temporal coverage of the in-situ data may impact the final mean estimate. Third, other kinds of observations could be used. In this study SAR derived velocity are used for validation purpose, but this kind of measurement could be included in the MDT inversion. In order to improve coastal area results, HF radar measurements are valuable, as Caballero et al. (2020)

show in the Bay of Biscay. They process HF radar velocities to extract only the mean geostrophic part and show that, when including this information in the MDT computation, the local mean circulation is better resolved. Finally, a realistic MDT


error should be estimated. A formal error is given as an output of the objective analysis. It highly depends on the parameters prescribed in input and it is known to be underestimated. We plan to use new method to estimate this field like for instance an ensemble method.

**Data availability**


The MDT CNES-CLS18 is available on AVISO+ website ([https://www.aviso.altimetry.fr/en/data/products/auxiliary-products/mdt.html](https://www.aviso.altimetry.fr/en/data/products/auxiliary-products/mdt.html)). Note that the CNES-CLS18 as described in this paper do not estimate MDT in the Mediterranean Sea. Though we have merged it with the SOCIB Mediterranean MDT (Rio et al., 2014b).

**Author contribution**


SM, MHR and HE have processed the data and computed the MDT CNES-CLS18. RH have computed the SAR currents in the Agulhas current area. CA, MC, HF, CP, PTS have analyzed and validated the MDT in their area of expertise (i.e. respectively Drake Passage, Australia, Gulf of Maine, Drake Passage, Chilean coast). GD and NP have supervised the study. SM and MHR prepared the manuscript with contributions from all co-authors.

**Competing interests**


The authors declare that they have no conflict of interest.

**Acknowledgements**

This study has been funded by the French space agency CNES. We thank the Beta users for their valuable feedbacks. Funding for the development of HYCOM has been provided by the National Ocean Partnership Program and the Office of Naval Research. Data assimilative products using HYCOM are funded by the U.S. Navy. Computer time was made available

by the DoD High Performance Computing Modernization Program. The output is publicly available at [http://hycom.org](http://hycom.org).

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
