# Peer review of "The new CNES-CLS18 Global Mean Dynamic Topography"

_Ocean Science, 2020_

## Author Response (AR1)

Dear Editor and referees,

In this document you will find answer to referees' comments. This document is organized as follow:

- 1. In italics: referees' comments
- 2. In **bold**: author's response and associated changes in manuscript.

**1. Answer to referee R1**

This very well written manuscript describes how the newest version of the CNES mean dynamic topography product is calculated, and is a straightforward read for those familiar with the 2009 and 2013 versions. As a user of the 2018 product already, I am happy to see this published in the peer-reviewed literature. The authors have produced a product of great value to the community, and I particularly enjoyed the case studies in Section 8 to validate the results and demonstrate the improved performance. Many of the results and figures in this section, such as Fig. 12, are very striking. As the authors note in their conclusions, many different factors went into these improvements; while one certainly wonders about the relative impact of improved techniques vs. higher data density/higher spatial resolution, in the end a user wants all of these improvements.

*I have only very minor comments, and recommend that the manuscript be published with only minor revisions.*

Thanks a lot for your positive comment and for your specific questions that will help us to clarify the manuscript.

94: there should be a citation or acknowledgement for the SD-DAC.

**We have added the reference to Lumpkin et al. (2013) at line 96.**

113: what does "section 0" mean? Should this say section 5?

**Yes indeed. We have corrected that.**

168: is there a reason why the undrogued drifters were not used for z=0m? Is this due to slippage (noted later in the manuscript)?

Exactly, we thus have added the following sentences at line 173: "Note that Argo floats were found to be much less affected by wind slippage compare with undrogued drifting buoys certainly thanks to their design (Rio et al., 2014a). Consequently, we do not need to correct Argo float drift from wind slippage and thus we can use Argo floats to estimate Ekman model at the surface (z=0) unlike undrogued drifting buoys that are affected by wind slippage (see section 5.2)."

170-172: no lowpass is applied to the Argo data, because the floats aren't at the surface long enough to allow for this. The authors should note that explicitly, and that these data thus include much more noise from high frequency motion. This again makes me wonder why the ~hourly undrogued drifter data wasn't used for z=0m. [NOTE: the authors address this on lines 265-268. I left this comment as a notice that readers may be wondering about this earlier.]

We have added an explanation line 172: "Unlike drogued SVP drifters, Argo float trajectories are not filtered since there is only one velocity estimation every 10 days." The undroggued drifters are too much affected by wind slippage to be used to estimate Ekman component. However, since the Ekman component is estimated at z=0m using Argo floats, then this component can be removed to estimate the wind slippage that affect undroggued drifters as described in section 5.2.

201-205: this is a great result! Very well presented.

**Thank you very much**

263: how does this work at very low latitudes? Wouldn't max(Pi, 24h) go to infinity?

We've forgot to say that in practice an upper bound is set at 6 days for the cutoff wavelength. We have added this information line 272: "Note that an upper bound is set at 6 days for the cutting period to avoid too high values at low latitudes."

Technical issues: 65, 178: misplaced (); 226: font size change.

Done, thanks.

**2. Answer to referee R2**

I agree on the comment of referee #1 that this is a well written manuscript that nicely describes the updated methodology to produce the MDT and presents striking improvements of the new MDT compared to older versions.

What I would like to see are comparisons with geodetic MDTs that apply new combined geoid models using GOCE data (XGM, SGG-UGM, GOCO05c, GECO). In I48 it is stated, that for spatial scales shorter than 100km other information than those provided by geodetic MDTs alone is needed. But actually, when applying combined geoid models clearly signal is found below 100 km scale. This has been shown at least for the Gulf Stream and the Kuroshio (Siegismund 2020). I'm pretty sure the CNES-CLS18 MDT resolves shorter scales than any geodetic MDT, but to show this would even underpin the additional value of the new CNES-MDT compared to any pure geodetic MDT.

Thanks a lot for your comment. We agree that geodetic MDTs computed from combined geoid model could resolve scale smaller than 100km. In the revised manuscript we precise

that we talk about "satellite only" geoid model (lines 45-53). However, to be independent from any apriori MDT, CNES CLS MDT do not use combined geoid models that use a priori MDT to convert altimetric information to gravimetric signal.

We agree also that using combined geoid model, geodetic MDTs may have valuable signal at scale shorter than 100km especially in main currents but in less energetic area residual noise could be important and dealing with this noise is not trivial.

We thus added a comparison with a well-known MDT: the DTU19 that we also mentioned in the introduction of the revised manuscript. Figure 7 have been updated to add the power spectra of the DTU19 MDT that is based on a combined geoid. We see on this figure that the CNES-CLS18 resolves smaller scales than the DTU19 MDT. We have added a comment on that line 330. We have done other analyses that we do not include in the paper to not make it too long and since they lead to the same conclusions:

We have looked at the associated velocities. FigureS1 shows a spectral analysis in the Argentine Bassin around the Zapiola drift. This figure leads to the same conclusion than Figure 7 of the revised manuscript: CNES-CLS18 resolves smaller scales than the DTU19 MDT. Even, the DTU19 MDT has the same spectral content than the CNES-CLS18 first guess. FigureS2 confirms the spectral analyses: the Glorys12 and the CNES-CLS18 MDTs resolves similar structure with similar intensity while DTU19 and the CNES-CLS18 first guess are smoother. The DTU19 is slightly more energetic than the CNES-CLS18 first guess.

Figure S1 : Spectral Power Density of the zonal geostrophic velocities associated with CNES-CLS18 MDT (black), its first guess (red) and DTU19 MDT (green)